# Shrinkage and Mitigation Strategies to Improve the Dimensional Stability of CaO-FeO_x_-Al_2_O_3_-SiO_2_ Inorganic Polymers

**DOI:** 10.3390/ma12223679

**Published:** 2019-11-08

**Authors:** Guilherme Ascensão, Glenn Beersaerts, Maurizio Marchi, Monica Segata, Flora Faleschini, Yiannis Pontikes

**Affiliations:** 1Italcementi S.p.A, HeidelbergCement group, via Stezzano, 87, 24126 Bergamo, Italy; m.marchi@italcementi.it (M.M.); m.segata@italcementi.it (M.S.); 2Department of Materials Engineering, KU Leuven, Kasteelpark Arenberg 44, 3000 Leuven, Belgium; glenn.beersaerts@kuleuven.be (G.B.); yiannis.pontikes@kuleuven.be (Y.P.); 3Department of Civil, Environmental and Architectural Engineering, University of Padova, Via Marzolo 9, 35131 Padova, Italy; flora.faleschini@dicea.unipd.it

**Keywords:** inorganic polymers, slag reactivity, curing conditions, shrinkage, building materials, mortars

## Abstract

Volumetric stability is an important aspect of the performance of building materials, and the shrinkage of CaO-FeOx-Al_2_O_3_-SiO_2_-rich inorganic polymers (IPs) has not been thoroughly investigated yet. Hence, this paper describes the outcome of a study conducted to investigate ways to minimize their shrinkage using different curing regimes. Two different slags were used as case studies to assess the robustness of the developed mitigation strategies. IP pastes and mortars were cured at (i) room condition, (ii) in slightly elevated temperature (60 °C for 2 d) and (iii) in a water-saturated environment. The reaction kinetics and formed products were examined on IP pastes, while mortars were made to characterize the 28 d pore structure, autogenous shrinkage, drying shrinkage, and strength development. The results showed that the precursors’ reactivity and curing conditions severely affect shrinkage mechanisms and magnitude. Volumetric changes in the plastic stage can be related to the precursors’ reactivity but drying shrinkage was the driving mechanism affecting the volumetric stability of all IP mortars. Understanding the effect of a precursor’s composition and curing conditions on shrinkage is fundamental to develop proper mitigation strategies and to overcome one of IPs’ main technical drawbacks.

## 1. Introduction

Like many other sectors, cement and construction industries are gradually transitioning towards more efficient and sustainable resource management, greener (lower environmental-footprint) and circular economy models. In fact, over the last decades, these industries have already made tremendous effort to reduce their environmental impact, most notably greenhouse gases emissions, by, namely, (i) improving kiln efficiency and reducing electricity usage; (ii) using alternative feedstock materials as fuel in manufacturing (e.g. municipal solid waste and refuse-derived fuel); (iii) replacing non-renewable aggregates with recycled materials or industrial by-products; and (iv) by using supplementary cementitious materials to partially replace ordinary Portland cement (OPC) [1,2]. Complementary to the above actions that aim to reduce the environmental impact of OPC and OPC-based building elements, the development of alternative binders have regained attention. A great deal of research has been performed to develop alternative binders and to overcome their technical challenges. Promising alternative binders comprise of calcium sulfo-(iron)-aluminate cement (CSA), blended, aether and celitement cements (as indicative examples), and alkali-activated materials (hybrid cements and inorganic polymers).

In the field of inorganic polymers (IPs), a distinction should be made between binders resulting from calcium-(iron)-aluminosilicate precursors, from here on simply called IPs, and a particular subgroup of binders produced from low-calcium aluminosilicate precursors called geopolymers (GPs). Despite the chemical differences between GPs and IPs, these materials are reported to have excellent mechanical strength properties [3], high chemical resistance (acidic or sulfate environments) [4,5], and considerable resistance to high-temperatures [4,5]. Apart from applications in the construction sector, GPs and IPs had shown promising potential in a vast group of novel applications that include, among others, wastewater decontamination [6], pH regulation [7], membrane and membrane supports [8,9], catalysts, catalyst supports and scaffolds [10], high-temperature processes and acoustic insulation [11], radioactive waste management [12], and space construction [13].

Nonetheless, GPs are synthesized from valuable raw materials, such as metakaolin and class-F fly ash, and in addition to the cost of the activating solutions, the economic viability is challenged [14,15]. On the other hand, IPs can be synthesized from a wider source of low-grade raw materials, such as calcined clays, volcanic ashes, metallurgical residues, high-temperature processed waste, or industrial residues [16]. Some of these secondary raw materials can be highly reactive, also allowing the reduction of the silicate/hydroxides dosage in the activating solution and/or permiting the use of inexpensive non-corrosive activators such as Na_2_CO_3_, Na_2_SO_4_ or their K-equivalents, adding both economic and environmental benefits [17].

Despite the latter considerations, IPs are still slowly being deployed as a commercial complement to traditional cementitious materials. Apart from factors relating to logistics, the supply chain, and the standardization, one factor that limits the large-scale implementation of IP mortars is their high susceptibility to shrinkage [18,19], which has been shown to be 2 to 4 times higher compared to OPC mortars [20,21]. The shrinkage of IP mortars can be mainly attributed to two different processes: autogenous and dry shrinkage. In open conditions, these processes occur simultaneously, but the mechanisms involved are rather different. Autogenous shrinkage can be defined as a physico–chemical phenomenon resulting from chemo–mechanical and hygro–mechanical interactions [22]. The former results from the difference between the absolute density of the reaction products and the starting materials (also known as chemical shrinkage), while the latter is driven by the progressive emptying of the initially saturated pore structure as the reaction progresses (also known as self-desiccation). Self-desiccation may generate considerable tensile stresses at the menisci, resulting in a severe inward contraction of the pore structure. These hygro–mechanical processes are rather analogous to drying shrinkage phenomena occurring when water leaves the system—through evaporation—generating tensile stresses, which causes further volumetric contraction. The aforementioned shrinkage mechanisms can be easily linked to the shrinkage behavior of traditional cement-based mortars; yet hydraulic and IP binders chemistries are entirely different which leads to incompatibility of most commonly used superplasticizers and shrinkage reducing agents with such novel materials [23]. Moreover, existing literature shows that shrinkage mechanisms and their magnitude in alkali-activated materials are heavily compositional dependent. Lee et al. [19] showed that shrinkage characteristics are primarily dependent on the precursors and the activating solution provided, and from the reaction products made thereof.

Considering the expected worldwide production growth of non-ferrous metallurgic slags and vitrified residues originated from thermal treatment of municipal wastes, the alkali activation of these CaO-FeO_x_-Al_2_O_3_-SiO_2_-rich slags has been considered as a possible large-scale valorization route. The implementation of such upscaling strategy will require the development of alkali-activated materials compliant with construction codes and standards where low shrinkage values are compulsorily prescribed. Yet, the existing literature on the shrinkage behavior of iron-rich IP pastes and mortars is surprisingly scarce [5]. Considering that shrinkage mechanisms can be inferred by reaction mechanisms and the formation of different binder structures [24], this study investigated the shrinkage mechanisms (and their magnitude) affecting CaO-FeO_x_-Al_2_O_3_-SiO_2_-rich inorganic polymers. For that scope, two CaO-FeO_x_-Al_2_O_3_-SiO_2_-rich precursors were used as case studies to assess the shrinkage behavior of iron-rich IP mortars. In addition, the effect of curing conditions on shrinkage behavior has been mainly examined in low-iron systems [25,26] and remains mainly unknown in iron-rich systems. The possibility of minimizing autogenous and drying shrinkage of Fe-rich IP mortars using different curing regimes was evaluated and the effects on the pore size distribution and strength development of Fe-rich IP mortars described.

## 2. Experiments

### 2.1. Materials

Two vitreous slags were used as precursors for IP binders. The slags were engineered to present chemical compositions representative of common non-ferrous metallurgy slags (KO slag) and vitrified residues originating from thermal treatment of municipal wastes (PS slag). The detailed production process of each slag can be found in [16] and [27]. The main components of both slags are SiO_2_, Al_2_O_3_, CaO, and FeO_x_, the main difference among them is the FeO_x_/CaO molar ratio. Both slags were received as granules and they were dried and milled before use. Commercial normalized sand (M31, Sibelco, Antwerp, Belgium; 99.5 wt.% SiO_2_) was used as aggregate and its particle size distribution is given in Figure 1. An activating solution, with a SiO_2_/K_2_O molar ratio of 1.60 and 70.0 wt.% H_2_O was prepared by dissolving potassium hydroxide beads (reagent grade, 85.0%, Honeywell, Belgium) in a commercial potassium silicate solution supplied by Silmaco, Belgium (SiO_2_/K_2_O molar ratio of 3.20 and approximately 60.0 wt.% H_2_O) and demineralized water. The solution was prepared at least 24 h in advance to allow it to cool down prior to IP paste and mortar preparation.

### 2.2. Methods

X-ray fluorescence (Bruker AXS S8 Tiger spectrometer, Bruker Italy S.r.l., Milano, Italy) was used to determine the bulk chemical composition of each powdered slag. The Fe oxidation state of each slag was quantified by performing a H_2_SO_4_-HF-H_3_BO_3_ treatment in an argon atmosphere, followed by Ce^4+^ titration as described by Close et al. [28]. The crystallinity of slags and IP pastes were assessed by X-ray diffraction (XRD), carried out on a conventional Bragg-Brentano Bruker D8 Advance diffractometer equipped with a Lynxeye detector (Cu Kα radiation λ = 1.54059 Å, divergence slit 0.5°, Soller slit set 2.5° + 2.5°, 5°–70° 2θ, step/size 0.02° and t/step 0.04 s.) and phase identification by EVA software (database ICDD-PDF-2, 4.2, Bruker Italy S.r.l., Milano, Italy). The particle size distribution of each slag was determined by a laser particle analyser using a dry method (Malvern Mastersizer 3000). The surface area of each powdered slag was determined via nitrogen adsorption/desorption methods (NAD) [29] and the Blaine method, according to the EN 196-6 [30]. NAD experiments record the adsorbed/desorbed nitrogen quantity as a function of the pressure imposed, allowing a deeper characterization of the structure of the powdered slags. Different evaluation methods—such as standard isotherms (t-plots), Langmuir, Brunauer-Emmett-Teller (BET) and Barrett-Joyner-Halenda (BJH) [31]—have been proposed to evaluate NAD SSA values; the T-plot method was used herein. 

The generated heat flow in inorganic polymeric (IP) pastes was determined by isothermal conduction calorimetry (ICC) at 20 °C. The ICC measurements were conducted in a 3116-1 TAM 83 Air (TA Instruments, New Castle, DE, USA) isothermal calorimeter using an external mixing procedure. The slags (10.0 g) and the activating solution were stored for 24 h at 20 ± 0.5 °C prior to the experiment to minimize heat turbulence. Afterward, the activating solution and the precursors were mixed for 1 min using an external mini-blender and introduced into the calorimeter. Each mixture was replicated once to guarantee data repeatability, and the average values are reported here.

The synthesis procedure of IP mortars involved: (i) introducing the slag to the activating solution and mixing (in a Hobart N-50 mixer) at low speed (139 rpm) for 60 s; (ii) adding quartz sand aggregates to the blend and mixing for another 60 s at the same speed; and (iii) mixing 30 s at high shear (285 rpm). After mixing, the mortars were cast into 4 × 4 × 16 cm^3^ stainless steel molds, sealed with plastic film and cured for 24 h at room temperature (20 ± 0.5 °C and 65% ± 5.0% relative humidity). After curing, the samples were demolded and kept under different curing conditions as described in Section 2.3.

The chemical structure of the IP pastes and the slags were analyzed using the attenuated total reflectance-Fourier tansform infrared spectroscopy (ATR-FTIR) (ALPHA 2 FTIR spectrometer, Bruker) method. For this purpose, 2 × 2 × 2 cm^3^ paste samples were produced following the procedure previously described but without the addition of aggregates to the blend. Prior to each test, the IP samples were ground with mortar and pestle. The spectra were analyzed in the reflectance transmission mode with a resolution of 2 cm^−1^.

The bulk density of IP mortars was determined by the relation between the weight and volume of each sample; their flexural and compressive strength was determined using a Universal Testing Machine (Instron 5985e) according to the EN196-1:2016 [32]. At least three samples for each formulation, curing conditions, and curing age were tested.

The autogenous shrinkage of IP mortars was investigated according to ASTM standard C1698 [33]. Two replicates of each mortar formulation were produced and used to fill corrugated polyethylene tubes. After placed in the apparatus, the length of the corrugated tubes was continuously recorded and used to determine autogenous shrinkage as a function of time. To evaluate the total shrinkage, IP mortars (4 × 4 × 16 cm^3^) were prepared with a metallic spindle (type I), according to EN 12617-4:2002 [34]. A minimum of two samples per each formulation and curing conditions were produced. The length of each mortar sample was monitored up to 56 days after casting by using a dial gauge. Weight variation was recorded during the same period.

The pore size distribution of each IP mortar was investigated by mercury intrusion porosimetry (PASCAL 140, Thermo Scientific). Unharmed samples from mechanical tests were collected and dried in a vacuum chamber for 5 hours (45 °C) prior to testing. A mercury surface tension of 0.48 N/m and a contact angle of 141.0° were set for the mercury intrusion porosimetry measurements (MIP).

### 2.3. Experimental Conditions

All mortar mixtures were designed to present a constant solution/slag mass ratio of 0.38 and a slag-to-aggregate mass ratio of 1.50 (Table 1); the molar ratios of the starting binders are shown in Table 2. Three curing conditions were imposed on the different slag-based IP mortars to assess the impact of slag characteristics and curing environment on the shrinkage mechanisms. The selected curing conditions comprised: (i) ambient curing, (ii) slightly elevated temperatures (60 °C for 2 d), and (iii) a 100% relative humidity (RH) environment at room temperature. The described conditions exclude the initial 24 h period where all samples were kept in the molds following the conditions described in Section 2.1. After demolding, ambient cured IP pastes and mortars were exposed to room temperature (20 ± 0.5 °C) and 65% ± 5.0% humidity. These experimental conditions allowed the determination of the IP mortars’ total shrinkage during curing and replicated the most likely environment in common construction applications. The second batch of specimens was thermally treated for 48 h at 60 °C after demoulding. The samples were sealed with a plastic film during thermal treatment to prevent severe drying. Such conditions guarantee that water evaporation is avoided, and given that plastic shrinkage already took place during the initial 24 h of curing (before demolding), it can be assumed that autogenous shrinkage is exclusive during the thermal treatment. It should be noticed that slightly elevated temperatures are known to increase the precursors’ degree of reaction and to accelerate the reaction kinetics resulting in the formation of different reaction products and porous structures [25,35,36,37]. Therefore, the results gathered will describe these combined effects preventing direct comparison with autogenous shrinkage processes occurring in IP mortars cured at room temperature. Notwithstanding, such curing conditions were imposed to assess if an increase of reaction kinetics and a faster consolidation of the polymeric network was able to sufficiently enhance strength development to withstand the tensile stresses generated onwards and increase the volumetric stability of IP mortars over time. The autogenous shrinkage of IP mortars cured at room temperature was determined by using corrugated tubes. Under this type of condition (room temperature and absence airflow), no volumetric phenomena are expected to occur other than chemical shrinkage (during the plastic stage) and self-desiccation as the reaction progresses. Direct comparison with shrinkage values registered in room-cured IP mortars was expected to allow the determination of the predominant character of drying or autogenous shrinkage.

The third batch of specimens was cured in a water-saturated environment. Similarly to corrugated tubes measurements, it is assumed that no drying shrinkage takes place. Since plastic shrinkage already occurred during the first day of curing, these trials intended to confirm the magnitude of autogenous shrinkage in IP mortars.

## 3. Results and Discussion

### 3.1. Precursors’ Characterization

The PS and KO slags’ chemical composition presented comparable amounts of SiO_2_ but very distinct contents of FeO_x_ and CaO (Table 3). PS slag presented a FeO_x_/CaO bulk ratio of around 1.0 while such ratio in the KO slag was approximately 13.0. The Al_2_O_3_ content in the PS slag was slightly higher than in the KO slag but the overall SiO_2_/(Al_2_O_3_+FeO_x_+CaO) molar ratio was lower, i.e., 0.82 and 1.24 for PS and KO slags, respectively. In low-Al_2_O_3_ systems, it is particularly important having the latter ratio in consideration since calcium and iron compounds have a remarkable influence on reaction kinetics and degree of crosslinking of the final polymeric structure [24,38,39]. Lastly, the iron phases were found to be predominantly in the bivalent oxidation state in both slags, being 92.0 % and 94.0%, in the PS and KO slags, respectively. 

Along with the slag bulk oxide composition, the particle size distribution (PSD) and specific surface area (SSA) are key parameters influencing the precursors’ dissolution rate and degree. For a given precursor, alkali-activation solution and precursor/solution ratio, PSD and SSA, control the activating solution sorption process and ensuing dissolution and gel formation. Hence, such physical parameters considerably influence reaction kinetics and products formed; thus, ultimately determining the properties of IP binders and mortars made thereof (e.g., shrinkage).

Figure 1 shows the slags’ particle size distribution. KO slag has a higher content of smaller particles but both precursors were characterized by a predominantly bimodal distribution with all particles being smaller than 0.1mm. SSA values determined by the Blaine method were found to be 5500 ± 400 and 4500 ± 200 cm^2^/g, respectively, for KO and PS slags. In NAD experiments, the absorption volume showed a linear dependence on the absorbed thickness, in which the SSA of each slag is given by the slope of the T-plot in Figure 2.

The two different slopes show that the SSA of KO slag was higher than for the PS slag, i.e., 1.0049 and 0.5470 m^2^/g, respectively. By confronting these results with SSA values obtained through the Blaine method, it can be seen that SSA values are systematically higher (Table 4). The explanation for such discrepancy can lie on the fact that NAD methods were originally developed to determine the SSA of predominantly flat-surfaced materials under the assumption of pores with cylindrical geometry and due to the porosity of slag particles itself. Nonetheless, both methods indicate that the KO slag has a larger internal accessible area, further confirming laser scattering and Blaine measurements. Lastly, the KO slag has a higher density than the PS slag, i.e., 3.41 and 2.97 g/cm^3^, respectively (Table 4). Such a difference is probably related to the much higher iron content of the KO slag.

### 3.2. Isothermal Conduction Calorimetry Testing (ICC)

Figure 3 shows the heat evolution rate and cumulative heat evolved of the alkali-activated KO and PS pastes. The reaction kinetics in alkali-activated materials (AAMs) are often characterized by two, and sometimes three, exothermic peaks depending on the used precursor and synthesis conditions [36]. The first exothermic peak (Peak I) appears immediately after mixing the powdered precursor with the activation solution. This peak is attributed to wetting and instant sorption of the activating solution on the precursors’ surface and ensuing dissolution reactions. Wetting, sorption and dissolution processes can be prompted by rising precursor SSA, and therefore increased magnitudes could be expected as SSA rises (KO > PS). Figure 3a shows the heat evolution rate during the initial 30 min of reaction. As can be seen, due to the instantaneous nature of Peak I, employing ICC with pastes’ external preparation did not allow the determination of the maximum heat release. However, the initial isothermal data seem to suggest that PS pastes present superior sorption and initial dissolution. These results indicate that precursor dissolution and isothermal data cannot be solely attributed to precursor SSA, but also reflect the difference in the slags’ density and reactivity. In fact, both precursors are complex glasses where network formers and modifiers influence the degree of structural disorder and therefore their reactivity towards an alkali solution. Moreover, it is still not consensual whether the Fe species present in glassy phases act as network formers or network modifiers, further increasing the complexity and uncertainty concerning the dissolution mechanisms involving Fe-rich precursors. The complex chemistry of the slags did not allow the clear identification of the individual effects of single oxides, but the very early stage of reaction seems to be mainly governed by the precursors’ chemistry instead of their physical characteristics (SSA). After Peak I, both pastes presented the characteristic deceleration period followed by a second exothermic peak (Peak II). Peak II is commonly attributed to the oligomerization and polymerization stage. The magnitude of Peak II and the time corresponding to its maximum varied notably according to the precursor used. KO pastes presented a much broader, smaller and delayed Peak II compared to PS (Figure 3b). In a first attempt, some parallelisms can be drawn from the simplified and widely described case of alkali-activated metakaolin pastes (MK), even though the chemistry of the precursors used in this work is more complex. In this system, the reaction starts with the hydrolysis of T–O (T = Al, Si) bonds followed by the polymerization of initially Al-rich phases, which will be progressively converted into more Si-rich phases as the reaction continues [5]. Despite the most likely structural differences existing between MK, KO, and PS systems, the slightly higher availability of Al species in PS pastes compared to KO ones was enough to expect a shorter time required to reach the supersaturation level and to initiate polymerization. In addition, both precursors used in this work presented significant Ca and/or Fe contents which further complicate the description of the kinetics. Van Deventer et al. [38] reported that Ca^2+^ and Fe^2+/3+^ cationic species rapidly dissolve and precipitate, providing a large number of extra potential nucleation sites. The accelerating effect of higher Ca amounts can result in the removal of OH^−^ ions from the solution which has a net effect of lowering the solution pH. A decrease in the solution pH will affect further dissolution/precipitation rate and reduce the solutions supersaturation levels. These competing effects hamper the definition of a general reaction kinetic model, but Salman et al. [40] observed in their work very early peaks and concluded that it may be related to the enhanced reaction kinetics due to the hydrolysis of Si–O–Ca and Ca–O–Ca linkages. KO pastes presented a second exothermic peak which is much broader and smaller compared to PS pastes, which indicates that when the FeOx/ CaO ratio increases, the effect of OH^−^ ions removal may prevail (Figure 3b,d). Daux et al. [41] showed that, under slightly alkaline conditions, the precipitation of dissolved Fe was much faster than Si and Al, which could lead to a considerable pH decay in KO pastes, resulting in a slower initial reaction progression. In fact, a higher cumulative heat release in PS pastes can be seen at early age (Figure 3d). Yet, it was later observed that after the initial 24 h, the heat released in KO pastes was still considerably high, while in PS pastes it was less significant. This may indicate a higher precursor dissolution degree, which is consistent with the increased mechanical performance exhibited by KO mortars cured at room temperature (see Section 3.7). As both precursors were composed of approximately the same SiO_2_ bulk content, extending the reaction may promote the formation of more silica-rich gels which are known to have a significant contribution to strength development [42]. It should be mentioned that the reaction of both KO and PS pastes were not completed after 550 h since the cumulative heat curves still present positive slopes. The ongoing reactions can still cause microstructural rearrangements and have an impact on the strength development of IP pastes and mortars. Lastly, a small third exothermic peak was identified only in PS pastes, appearing approximately after 72 h (Figure 3c). This third peak suggests a residual reorganization or crystallization at later stages in PS pastes [36] but further work is needed to shed light on the precise mechanisms involved here.

### 3.3. XRD Analysis

Figure 4 shows the XRD patterns of the precursors used in these experiments and the IPs pastes made therefrom. PS slag patterns exhibited a pronounced hump between 20°–40° 2θ confirming its predominantly amorphous nature. In the KO slag, such a hump was less visible, probably due to the considerable content of iron that increases the patterns’ background baseline. Still, the absence of significant crystalline phases in both slags revealed their glassy nature, thus suggesting high solubility towards an alkali medium. These observations agree with ICC data, where a significant exothermic reaction took place immediately when the alkaline solution was introduced to the precursors.

When comparing precursor and paste patterns, it is evident that the produced binders mainly conserve the structure of their parent precursor (Figure 4). No significant crystalline phases were formed in room-cured samples (65% RH) after 28 days. The absence of carbonated products suggests a low susceptibility of both binders to atmospheric carbonation, which however should be experimentally proved. Moreover, the absence of carbonation products in heat-treated pastes was also observed. As those pastes were cured for 2 d in a sealed environment and, therefore in saturated conditions, elevated RH levels do not seem to promote higher levels of carbonation. In addition, the ATR-FTIR results discussed in the following section show that all IP pastes presented similar peak attributed carbonated groups, which in turn suggests the formation of similar XRD amorphous structures. Therefore, despite the XRD patterns of pastes cured in a saturated environment for 28 d not being provided here, a significant formation of carbonation products was not expected in those samples.

### 3.4. ATR-FTIR Analysis

Figure 5 shows the IR spectra of PS and KO pastes cured under different conditions. Despite the different curing conditions, all the produced pastes presented similar peaks, albeit with different shape, intensity, and position (Table 5). The three main peaks correspond to: (I) the T–O (T = Al, Si) bonds stretching and out-of-plane bending vibrations, also known as rocking band, at 440 cm^−1^; (II) Si–O bonds in-plane bending vibration at 700 cm^−1^; and (III) the asymmetric and symmetric vibrations of Si–O–T bonds, also known as stretching band, at 970 cm^−1^ [27,28,43].

Some minor shifts in the location of the rocking band peaks were visible but no systematic correlation was depicted between its location and the precursors chemistry or curing conditions studied here. In the case of Fe-rich silicates, the band corresponding to in-plane bending appears to overlap with a characteristic band of iron oxide phases located around 600–700 cm^−1^ [44]. This overlap makes it difficult to attribute this peak to Si–O in-plane bending or to the vibration of Fe–containing units. On one hand, the location of this peak remained roughly unchanged (≈ 700 cm^−1^) in all IP pastes. On the other, the location of stretching band peak (≈ 900 cm^−1^) was clearly influenced by the imposed curing conditions. When comparing KO pastes cured in different conditions, it is interesting to see that the minimum value of the stretching band for the room-cured samples was located at the lowest wavenumbers. The relocation of this peak to higher wavenumbers suggests that heat treatment and curing in a steam-saturated environment promoted the formation of more polymerized reaction products as a completely polymerized 3-dimensional glass will have the Si–O–Si stretching band around 1100 cm^−1^ [45]. These results can be explained by the increase in the reaction degree of KO slag, leading to the formation of more Si-crosslinks, and/or to iron oxidation processes. Despite the oxidation mechanism of iron is not been fully understood yet, the work of Peys et al. [24] demonstrated that oxidation reactions are not driven by atmospheric air since it still happens in inert environments. In their work, the connectivity of the silicate network does not seem affected by oxidation reactions nor suffer modifications due to late age oxidation. Curing conditions may have affected the incorporation of Fe^3+^ species into the silicate networks and hence changed the amount of available non-bridging oxygens in the system, which can contribute to an increase of the degree of polymerization in the formed structures. Nonetheless, the formation of more polymerized structures is in agreement with mechanical strength results discussed later in this work where IP mortars cured at room conditions presented the lowest flexural and compressive strength. A similar effect was only observed in the FTIR spectra of PS pastes cured in a saturated environment. In all IP pastes, four other bands were formed. The broad peak between 2500–3700 cm^−1^ and the smaller peak around 1630 cm^−1^, corresponding to the H–O–H bonds deformation and O–H groups stretching vibrations, respectively. The use of different precursors and curing conditions had a minor impact on these peaks, being only visible a slight decrease of intensity in the peak located at 2500–3700 cm^−1^ on heat-cured pastes. This can indicate a small reduction of the amount of H_2_O molecules present in these samples. A small peak between 1400–1450 cm^−1^ is present in all IP pastes spectra but less pronounced in heat-cured samples. A fourth peak, located around 855 cm^−1^, appeared in pastes cured at room conditions and in a water-saturated environment. The latter two peaks can be attributed to the O–C–O stretching vibrations from carbonate groups. The formation of such carbonate groups is often attributed to the reaction of alkali cations with atmospheric CO_2_ [40,43].

### 3.5. Shrinkage Behavior of IPs Mortars

#### 3.5.1. Autogenous Shrinkage

Figure 6 shows the autogenous shrinkage of KO and PS mortars. PS mortars presented high autogenous shrinkage after 7 days (≈−2.9 mm/m) being several times higher than those reported in OPC systems and slightly above those reported in BFS-fly ash alkali activated mortars [19]. Possible explanations for this high autogenous shrinkage can be the large chemical shrinkage, the high saturation degree of the porous structure, the poor stiffness and the fine pore structures of alkali activated materials [18]. After the initial stage of reaction, KO mortars have a very distinct autogenous shrinkage behavior. Approximately 8 h after mixing, KO mortars started to exhibit a considerable expansion behavior which overcomes the initial plastic shrinkage. It is interesting to notice that the inflection point on KO shrinkage coincided with the heat flow peak (Peak II) detected in ICC measurements. Secondary exothermic peaks are associated with polymerization reactions, which may suggest that the observed expansion is related to the polymeric binder skeleton formation and ongoing reactions during that period. Autogenous expansion has not been broadly reported in AAMs but the works of Ascensão et al. [39], Soliman et al. [46], and Sant et al. [47] have shown that some shrinkage reducing agents can induce early age autogenous expansions. Sant et al. [47] attributed this expansion to crystallization stresses generated due to the supersaturation of the pore solution with portlandite phases. A similar supersaturation with portlandite phases seems unlikely in KO mortars, but the considerably high iron content of the binding phases in those mortars may have interfered with crystallization processes, possibly partially explaining the observed expansion. Other possible explanations can include the formation of ettringite due to the presence of SO_3_ and the oxidation of the metallic Fe present in KO slag in the alkaline medium. Furthermore, Bumanis et al. [48], recently explored the used iron sulfite (FeSO_3_)-rich precursors to generate sulfur dioxide and create porous materials. The described foaming reactions typically occur in a short timeframe after mixing, implying that the contact between the metallic species and the alkaline medium could have been prevented during the initial hours of reaction. The formations of gaseous phases typically have severe repercussions on the microstructure of AAMs, increasing total porosity and reducing density and mechanical resistance. None of the latter was observed in KO mortars which suggests that the explanation for such expansive behavior may reside on modifications imposed on crystallization processes and on the binding phases formed.

The expansion was followed by a post setting period where self-desiccation imposed some volumetric contraction (Figure 6). Considerable differences in binder chemical shrinkage and mortar degree of saturation, stiffness and pore size distribution may also have further contributed to accentuating the differences between the autogenous shrinkage of KO and PS mortars; a more thorough analysis of the mortars’ porous structures is provided in Section 3.6.

#### 3.5.2. Drying Shrinkage and Weight Loss

Similar to cementitious materials, drying shrinkage in AAMs is related to water loss in the pores, resulting in significant capillary stresses. Common strategies to mitigate drying shrinkage include increasing the aggregates’ content in the mix design, and the use of shrinkage reducing agents (SRA) [20,49]. In this study, another shrinkage mitigation method was investigated, which is the effect of the curing regime at the initial stage of reaction on drying shrinkage. Bakharev et al. [26] studied the effect of curing slag-alkali-activated concrete at slightly elevated temperatures, and they showed that heat curing can effectively reduce the shrinkage of alkali activated concretes, reaching values comparable to OPC concrete. They also found that heat treatment history has an effect on drying shrinkage and recommended a pre-treatment period at room temperature before heat curing [26]. Thomas et al. [50] investigated the effects of different curing methods on drying shrinkage of alkali-activated fly ash and granulated blast-furnace concrete, reporting that heat curing could improve volumetric stability up to 75% in alkali activated concretes. The aforementioned studies suggest a consistent beneficial impact of heat curing and a similar procedure was carried out in this work. Furthermore, since drying shrinkage is a hygrothermal process and, therefore the dimensional stability of AAMs is sensitive to evaporation processes, IP mortars were also cured in a steam-saturated environment. These experiments intended to investigate if drying shrinkage was the main factor measured in room-cured mortars. The results of specific mass variation and total drying shrinkage of alkali-activated mortars cured in different conditions are shown in Figure 7.

Independently of the precursor used (KO or PS slag), the total drying shrinkage at room conditions mainly occurred in the initial seven days of curing, evolving afterward towards a plateau value. These results were consistent with the expectations based on existing literature [18,19,26] and the authors’ previous findings [39,51]. The drying shrinkage of PS mortars was larger compared to KO mortars, −7.9 and −7.1 mm/m, respectively. Isothermal data have shown more intense reactions in KO pastes at room temperature, which may indicate the formation of more binding phases. Despite no significant differences have been observed in FTIR and XRD results, the formation of more reaction products seems to be supported by the higher strength development of KO mortars at room conditions (see Section 3.7). These results are in agreement with the findings of Thomas et al. [50] who reported that stiffer alkali-activated concretes presented improved dimensional stability. The total porosity of PS mortars was found to be slightly higher compared to KO mortars (54.6 and 52.1 mm^3^/g) but the smaller average pore size in PS mortars may further contribute to increasing the capillary tensile forces generated, hence increasing shrinkage deformation at room conditions. The accompanying mass variation has shown to be a more gradual process that continued along the 56 days of curing. The mass loss followed a similar trend to drying shrinkage, being greater in PS mortars and occurring primarily in the initial seven days of curing (Figure 7a). From the above observations, it can be concluded that the highest volumetric stability of KO mortars cured at room conditions can be attributed to several complementary phenomena, including the amount of reaction products formed and the characteristics of their microstructures. When accounting for shrinkage/expansion occurred during the initial 24 h after mixing, that comprises plasticstate, the difference between the two mortars is further exacerbated. Considering the most extreme cases of autogenous shrinkage in the initial 24 h as reference values, the overall shrinkage of PS mortars was found to be approximately −9.8 mm/m after seven days of curing at room conditions (Figure 8a). On the other hand, the plastic expansion observed in KO mortars could partially mitigate the subsequent drying shrinkage, leading to an overall shrinkage of −2.5 mm/m for the same period (Figure 8b). It should be noticed that total drying shrinkage values comprise the contribution of autogenous processes occurring during the hardened state. It is interesting to mention the similarities between the shrinkage values of IP mortars cured in saturated environments, where the exclusive existence of autogenous shrinkage can be assumed, and the long-term autogenous shrinkage measured using corrugated tubes which confirms the less significant role of autogenous shrinkage on the mortars’ hardened state.

Thermal treatment was found to decrease the magnitude of drying shrinkage and accelerate the drying rate in both mortars. The water loss mainly occurred during the 48 h of thermal treatment and its total value was considerably lower than in room-cured samples. KO and PS mortars presented similar mass losses, 2.3% and 2.6% by mass, respectively. However, similar to room-cured mortars, a slightly higher mass loss was observed in PS mortars, which suggests a consistent correlation between the precursor’s chemical composition and samples’ weight loss. The formation of finer porous structures at slightly elevated temperatures was confirmed by MIP experiments (see Section 3.6) and the reduction of the mean pore size is often associated with the increase of shrinkage, due to an increase of the capillary tensile stresses generated [50]. Nonetheless, thermal curing promoted faster consolidation of polymeric structures and enhance strength development (see Section 3.7). The development of a stiffer solid network enables heat-treated mortars to accommodate higher tensile stresses generated during drying, reducing the total drying shrinkage by 30%. Similar to room-cured samples, heat-treated KO mortars presented superior volumetric stability if compared to PS mortars, as their total drying shrinkage was lower by 0.6 mm/m after 56 days of curing (Figure 7). These results show that, independently of the curing regime and modifications imposed on the pore structures, KO mortars presented enhanced volumetric stability relative to their PS counterparts (see Section 3.6). However, as such outcome is the interplay result of the many factors contributing to the slags’ reactivity, safe conclusions regarding the precursors’ chemical composition cannot be drawn. Moreover, when combining the autogenous and total drying shrinkage, one can see that heat-cured KO mortars present minimal shrinkage (−0.1mm/m), while thermal treatment was only able to reduce the shrinkage of PS mortars to −7.4mm/m.

Mortars cured under a steam-saturated environment exhibited higher dimensional stability along the testing period, which denoted the vital role of water evaporation on AAM shrinkage processes. In these mortar samples, shrinkage resulted exclusively from autogenous processes and roughly coincided with the results discussed in 3.5.1. Autogenous expansion of moist-cured mortars in the hardened state seems to be slightly affected by the precursor’s different reactivity. However, the reduced magnitude of these results makes it difficult to postulate such a conclusion. Moist-cured samples presented a slight increase of mass during the initial 14 days of curing which can be attributed to the partial filling of open pores.

### 3.6. Mercury Intrusion Porosimetry

The porosity of AAMs is known to largely influence their shrinkage processes and therefore the pore size distribution of PS and KO mortars were evaluated, and results are discussed here in detail. Pores can be characterized according to their size and grouped in four main categories: (i) micropores, <1.25 nm, (ii) mesopores ranging from 1.25–25 nm, (iii) macropores ranging from 25–5000 nm, and (iv) entrained and entrapped air voids, and microcracks >5000 nm [21]. Micropores are inherent to reaction products, whereas capillary pores (comprising both meso- and macropores) can be seen as the residual unfilled spaces between reaction products, aggregates, and binding phases. Drying shrinkage greatly depends on the capillary pore size distribution since it determines the stresses generated by water evaporation. Capillary stresses result from the difference between the atmospheric pressure at the meniscus and the internal pressure of the pore solution, as expressed by the Young–Laplace equation. Internal capillary pressures increases as the meniscus radii become smaller, increasing the forces exerted on adjacent pore walls, resulting in higher shrinkage strains. The tortuosity and connectivity of pore structures are also important parameters affecting water movements in porous structures and the tensile stresses generated.

Figure 9 shows the pore size distribution of room and heat-treated mortars. In general, MIP data show that mortars cured at room conditions have higher porosity values and a broader pore size distribution. The average cumulative pore volume of samples cured at room conditions was 54.6 and 52.1 mm^3^/g in PS and KO mortars, respectively. A systematic effect of heat curing was observed but more evident in KO mortars. Heat treatment reduced the pore volume of PS and KO mortars to 43.7 and 37.6 mm^3^/g, respectively; representing a 19.9% and 27.9% reduction. Thermal treatment also influenced the pore size distribution by decreasing the relative proportion of macropores, while promoting an increase in the volume of mesopores. A reduction of the macropores’ average size was visible in both mortars. Slightly elevated temperatures modify the reaction kinetics and increase the amount of reaction products formed [25,35,36,37], hence promoting the formation of finer and less porous structures. The higher amount of reaction products formed contributes to partially fillcapillary pores, thus reducing the total pore volume and the average pore size. Filling capillary structures also contribute to form higher amounts of closed pores, which are able to encase water in their structure and thus reduce evaporation. The formation of denser structures is in agreement with the results in Table 6, where heat-treated mortars presented the highest bulk density values. The formation of less porous microstructures also contributed to enhance the strength development of heat-cured mortars (Figure 9), as it will be discussed in the following section.

### 3.7. Mechanical Properties

Figure 10 shows the flexural and compressive strength evolution of IP mortars (evaluated at 28 and 56 days), that were subjected to different curing conditions. KO mortars cured at room conditions presented slightly higher compressive strength results when compared to PS mortars. In both room-cured mortars, compressive strength values tend to decrease with curing time, while flexural strength increases in PS mortars. A similar increase in flexural strength of metakaolin-based geopolymers cured at ambient temperature was reported by Kuenzel et al. [52]. A critical molar water ratio is suggested, in which a uniform physical contraction in the gel microstructure can cause an increase in flexural strength while further drying induces more shrinkage and resultes in a decrease of flexural performance. The definition of the critical molar water ratio is most likely related to the gel structures, which in this work is directly correlated to the slags’ reactivity. The increase of flexural strength observed in PS mortars may suggest that a critical molar water ratio is possibly reached around the 56th day of curing. In KO mortars such water molar ratio must have been achieved during the initial 28 days of curing, followed by a decrease in flexural strength. Further investigation should be performed to determine how the slags’ reactivity defines the time required to achieve such a critical molar water ratio but the results herein suggest that higher FeOx/CaO molar ratios in the precursors’ composition accelerate such phenomena. Calorimetric data shown in Section 3.2 demonstrated that long term reactions still occur and may contribute to the formation of reaction products. Yet, the continuous mass loss and drying shrinkage eventually overwhelmed this positive effect and the mechanical performance of room-cured mortars was decreased.

Moist and heat curing notoriously increased strength development of IP mortars but a significant reduction of samples mechanical performance at later ages was also visible. Thermal treatment accelerated the formation of rigid networks, reduced the magnitude of drying shrinkage and decreased the mortars’ total porosity. All these factors contributed to an increase in the mechanical performance of the IP mortars. The decay of mechanical performance at later ages is in line with the findings of Bakharev et al. [26], who reported that heat-cured samples underwent a significant decrease in mechanical performance over time. Nonetheless, KO mortars presented considerably more stable mechanical performances than PS ones, which suggest that precursors similar to KO slag may beneficiate the formation of more stable reaction products at slightly elevated temperatures.

In steam-saturated environments, the continuous water supply that acts as a reaction catalyst and/or as a transport medium in alkali activation, along with a near absence of drying shrinkage in these conditions resulted in IP mortars with high flexural and compressive strength (Figure 10). After 28 days of curing, moisture-cured mortars showed a flexural strength of 8.3 MPa and 10.7 MPa, and compressive strength of 59.3 MPa and 81.6 MPa, for KO and PS mortars, respectively. With the exception of PS mortars where a decrease in flexural strength was observed at later ages, the mortars’ mechanical features increased with curing time. According to the above results, it can be concluded that the deterioration of mechanical performance observed in room-cured mortars can be primarily attributed to drying shrinkage processes. A similar deleterious influence can be assumed for the initial 28 days of curing.

## 4. Conclusions

In this study, the effects of different curing regimes on the shrinkage of CaO-FeO_x_-Al_2_O_3_-SiO_2_-rich inorganic polymeric mortars were investigated and the main findings can be summarized as follows:The precursors’ reactivity and curing conditions severely affected shrinkage mechanisms and its magnitude in IP mortars.The volumetric changes that occurred during the binders’ plastic stage were defined by the precursors’ reactivity. In the case studies analyzed here, considerable autogenous shrinkage or expansion has been observed in IP mortars.At room condition in a hardened stage, regardless of the precursors’ bulk composition, drying shrinkage was identified as the governing mechanism affecting the mortars’ volumetric stability, whereas autogenous shrinkage was less significant.The characteristics of the precursors affected the reaction kinetics and porous structures formed, thus modifying the mortars’ pore size distribution and greatly influencing their drying shrinkage behavior.Thermal treatment promoted a decrease in porosity and the redistribution of pores to lower dimensions, but it was found to be an effective shrinkage-control strategy as it reduced the total shrinkage of IP mortars by 30%. As expected, in water-saturated curing conditions, drying shrinkage was found to be negligible and corresponded to the results of corrugated tube autogenous shrinkage in the hardened stage.Thermal and moist curing promoted higher volumetric stability and considerably improved themechanical features of IP mortars. Mortars with enhanced flexural (up to 10.7 MPa) and compressive strength (up to 81.7 MPa) were produced.

Although heat and moist curing regimes may not be of practical interest in ready-mix concrete applications, understanding how curing conditions affect the volumetric stability of mortars with different chemical compositions may facilitate the large-scale application of CaO-FeO_x_-Al_2_O_3_-SiO_2_ inorganic polymers in market segments where such conditions can be more easily replicated, e.g. precast applications.

Understanding the effects of curing conditions on shrinkage mechanisms of alkali-activated materials is also expected to contribute to the development of novel waste-based inorganic polymers, and by doing so, to the upcycling of largely abundant CaO-FeO_x_-Al_2_O_3_-SiO_2_-rich residues.

## Figures and Tables

**Figure 1 materials-12-03679-f001:**
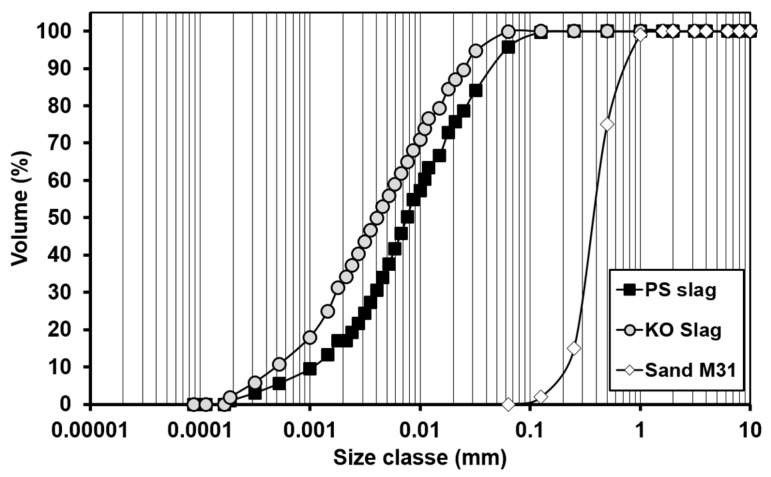
Precursors and aggregates particle size distribution.

**Figure 2 materials-12-03679-f002:**
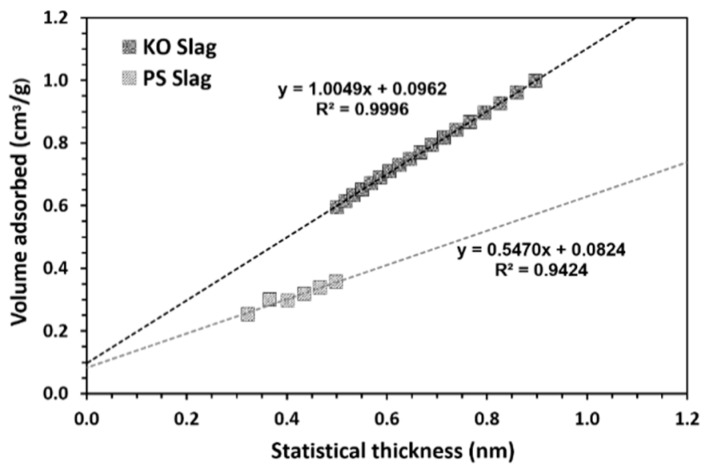
T-plot of inorganic polymers (IPs) precursors used: KO and PS slags.

**Figure 3 materials-12-03679-f003:**
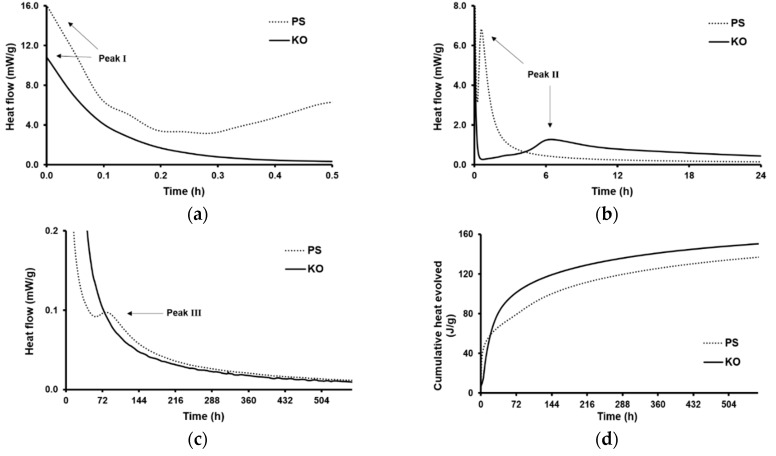
Isothermal calorimetry curves of alkali activated pastes. Heat evolution rate during the initial: (**a**) 30 min and (**b**) 24 h period while (**c**) shows a close-up of a secondary heat peak detected on PS pastes (Peak III). The cumulative heat released by PS and KO pastes is shown in (**d**).

**Figure 4 materials-12-03679-f004:**
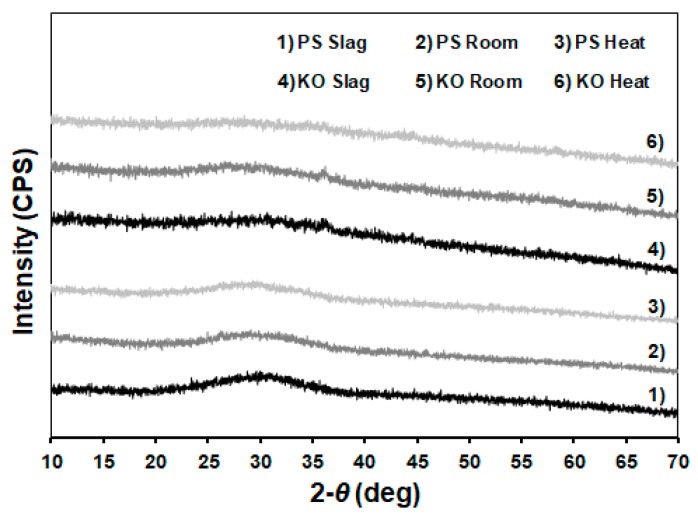
X-ray diffraction patterns of precursors (1 and 4) and alkali activated pastes made therefrom cured at room (2 and 5) and slightly elevated temperatures (60 °C for 2 d; 3 and 6). All the XRD patterns were collected after 28 days of curing.

**Figure 5 materials-12-03679-f005:**
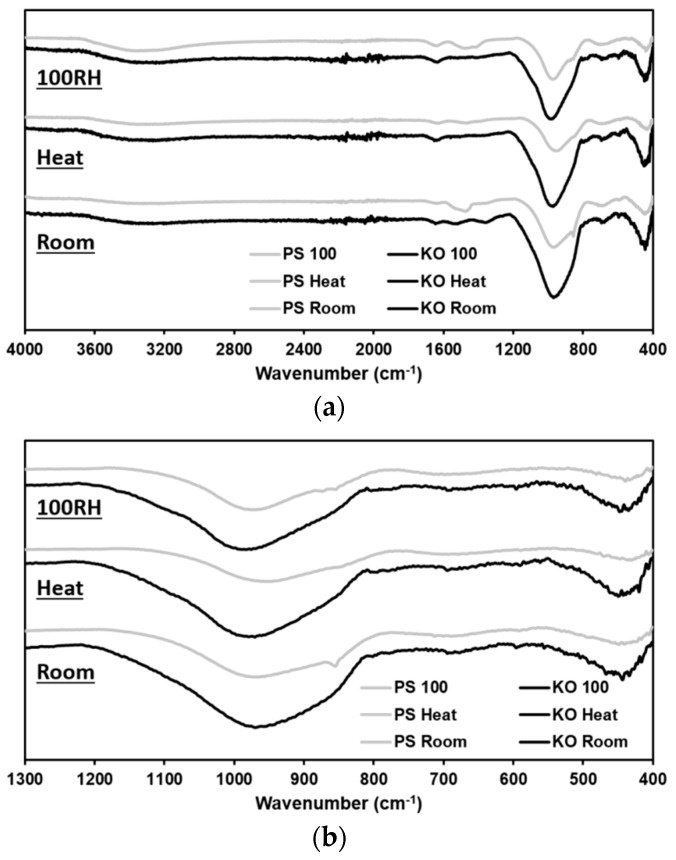
FTIR spectra of IP pastes cured under different conditions. (**a**) Full spectra collected; (**b**) close-up of low wavenumber region.

**Figure 6 materials-12-03679-f006:**
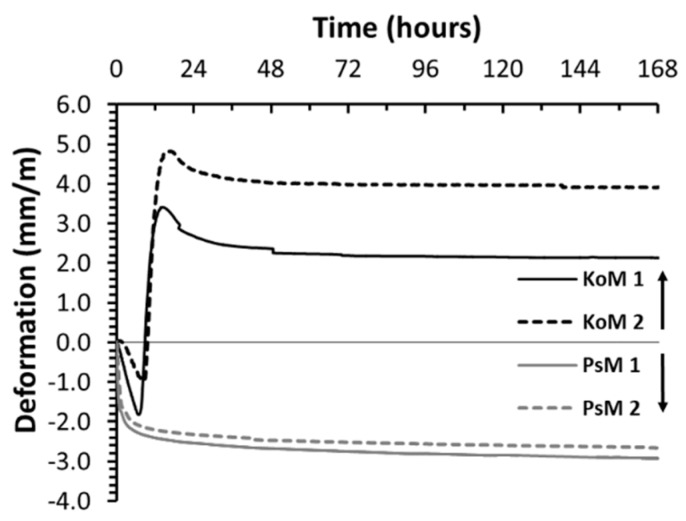
Autogenous shrinkage of alkali-activated mortars.

**Figure 7 materials-12-03679-f007:**
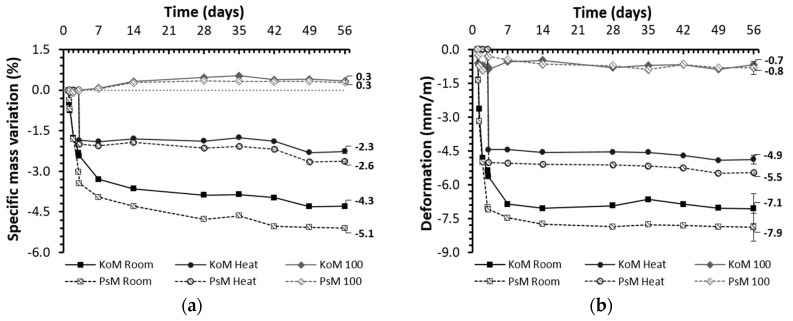
Specific mass variation (**a**) and total drying shrinkage (**b**) of alkali-activated mortars cured under different conditions.

**Figure 8 materials-12-03679-f008:**
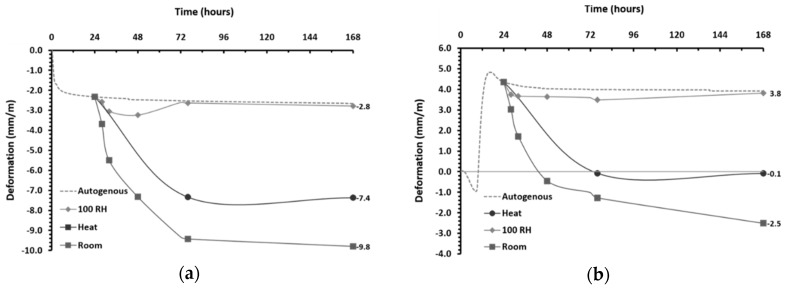
Combined representation of autogenous and total shrinkage of alkali-activated mortars cured under different conditions: (**a**) PS mortars and (**b**) KO mortars.

**Figure 9 materials-12-03679-f009:**
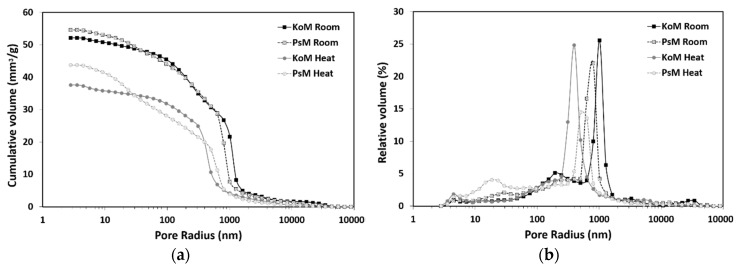
Experimental results of MIP measurements of room and heat-treated mortars: (**a**) cumulative pore volume and (**b**) relative pore size distribution.

**Figure 10 materials-12-03679-f010:**
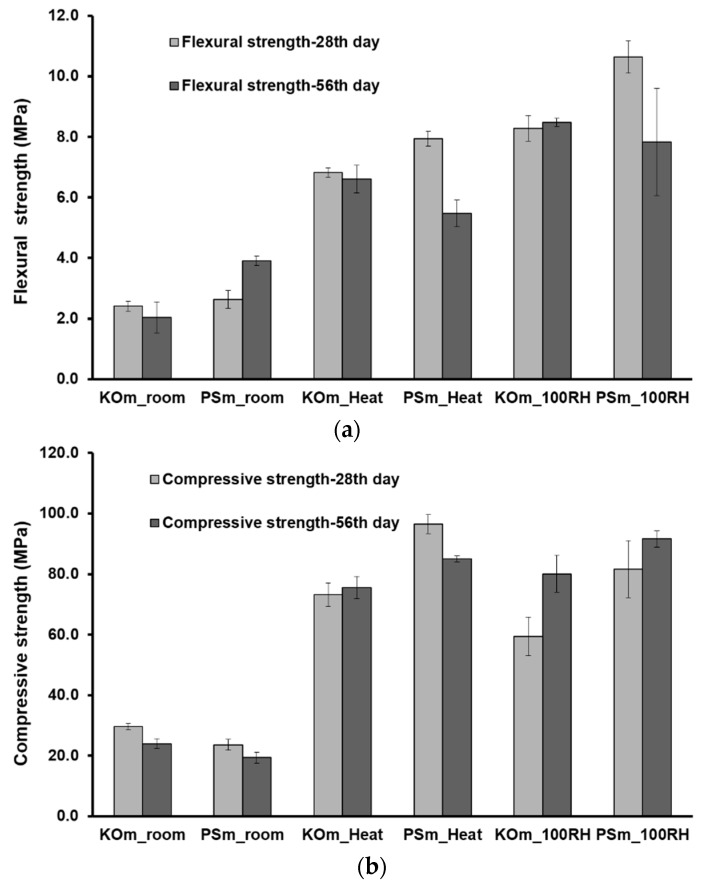
Flexural (**a**) and compressive strength of mortars (**b**) at different ages (28 and 56 days). Average results and corresponding standard deviations are provided.

**Table 1 materials-12-03679-t001:** Mortars mixture composition and slag-to-liquid and solid-to-liquid mass ratio.

Formulation	Mixture Portion (wt.%)				Slag/liquid	Solid/liquid
	PS	KO	Solution (aq.)	Agg.	(g/g)	(g/g)
**PS mortars**	34.72	-	13.20	52.08	2.63	6.57
**KO mortars**	-	34.72	13.20	52.08		

**Table 2 materials-12-03679-t002:** PS and KO binders’ molar ratios.

Formulation	Binders Molar Ratios		
	SiO_2_/Al_2_O_3_	M_2_O/SiO_2_	H_2_O/M_2_O
**PS pastes**	4.45	0.08	25.71
**KO pastes**	6.07	0.12	18.25

**Table 3 materials-12-03679-t003:** Representative oxide composition of the precursors used.

Oxide (wt.%)	PS_slag_	KO_slag_
Na_2_O	0.3	2.0
MgO	1.3	1.5
Al_2_O_3_	16.2	10.4
SiO_2_	34.9	34.8
P_2_O_5_	0.1	1.0
SO_3_	0.0	0.6
K_2_O	0.5	0.2
CaO	23.1	3.3
TiO_2_	0.6	0.3
MnO	0.1	0.9
FeO_x_	22.8	43.6
Loss on ignition	1.9	4.1
Fe^3+/^Fe_total_	0.08	0.06

**Table 4 materials-12-03679-t004:** Precursors’ density and specific surface area measured by Blaine and NAD methods.

Code	Density	Blaine	T-Plot
	(g/cm^3^)	(cm^2^/g)	(m^2^/g)
**PS slag**	2.97	4500 ± 200	0.55
**KO slag**	3.41	5500 ± 400	1.01

**Table 5 materials-12-03679-t005:** Location of main IR peaks in PS and KO pastes.

Code	Curing Cond	Peak Band I	Peak Band II	Peak Band III
		(cm^−1^)	(cm^−1^)	(cm^−1^)
**PS**	Room	448	712	970
	Heat	431	693	948
	Saturated	439	699	976
**KO**	Room	443	695	969
	Heat	450	695	970
	Saturated	448	693	983

**Table 6 materials-12-03679-t006:** Bulk density of IP mortars and corresponding standard deviations.

Code	Curing Cond	Bulk Density			
		1st day	4th day	28th day	56th day
		(g/cm^3^)	(g/cm^3^)	(g/cm^3^)	(g/cm^3^)
**PS**	Room	2.37 ± 0.02	2.29 ± 0.01	2.26 ± 0.01	2.25 ± 0.01
	Heat	2.39 ± 0.01	2.34 ± 0.01	2.34 ± 0.02	2.33 ± 0.02
	Saturated	2.38 ± 0.01	2.38 ± 0.01	2.38 ± 0.02	2.38 ± 0.02
**KO**	Room	2.39 ± 0.02	2.33 ± 0.02	2.29 ± 0.02	2.29 ± 0.02
	Heat	2.39 ± 0.01	2.35 ± 0.00	2.34 ± 0.02	2.34 ± 0.01
	Saturated	2.34 ± 0.01	2.34 ± 0.01	2.36 ± 0.01	2.33 ± 0.03

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
