# Peer review of "Shrinkage and Mitigation Strategies to Improve the Dimensional Stability of CaO-FeO_x_-Al_2_O_3_-SiO_2_ Inorganic Polymers"

_materials, 2019, doi:10.3390/ma12223679_

Round 1
Reviewer 1 Report
Report on manuscript Materials-632741, Ascensao et al.
General comments
This paper reports a well-designed and useful study of the effect of three different curing regimes on the properties of two fly ash starting materials of different compositions. Detailed attention is paid to the shrinkage properties, a factor that is often ignored in such studies, and a range of appropriate experimental techniques is brought to bear on these materials. One slight source of confusion for a reader is that both geopolymer pastes and mortars are reported, and it is not always clear which is being discussed, as in some places both these are simply called inorganic polymers (IPs) without differentiating between pastes and mortars. Precisely what is being discussed should be clarified, e.g. in the caption to Table 5 and elsewhere. The experimental procedure for producing the mortars is described (line 132) but not the pastes. The procedure for the latter should also be described.
The paper is written in generally clear English, but some ambiguities occur, as listed below.
Particular points for attention.
Line 57. A reference should be added for the use of the non-corrosive activators sodium or potassium carbonate or sulfate.
Line 138. FTIR analysis of the slags and the corresponding pastes are mentioned, but not the mortars. Were these also analysed?
Line 141. “grinded” should be “ground”
Lines 181, 436. The use of corrugated tubes to determine sample shrinkage mentioned here requires more explanation.
Table 1. Presumably the suffix m refers to the mortar, but this should be clarified. The composition of the IP paste, in terms of the molar ratios SiO2/Al2O3, M2O/SiO2 and H2O/M2O should also be given, either here or in another table, to enable the reader to relate the present IPs to other published results.
Line 302. This suggests that the XRD patterns of the pastes cured in steam were not obtained- is this the case, and if not a comment why not is needed. This sentence should be clarified.
Table 4. Looking at the FTIR spectra, the reporting of the peak positions to 1dp seems difficult to justify.
Line 378. The awkward construction of this sentence obscures the authors’ intended meaning. Clarification is needed.
Line 390. A reference is needed to the use of shrinkage reducing agents in geopolymers.
Fig. 9. Does this refer to the mercury porosimetry results? If so, it should be added to the figure caption.
Table 5. Are these results for pastes or mortars? This needs to be specified in the caption- to simply call them IPs could mean either.
Line 536. “undergone” should be “underwent”
Line 569. These conclusions seem to confuse mortars with IPs and the terms used interchangeably. To assist the reader, the terms pastes and mortars should be specified, both here and throughout the paper.
In summary, this paper is a useful addition to a little-explored region of the inorganic polymer literature, and provided these points are addressed, it should be published.
Reviewer 2 Report
The manuscript from Ascensao and co-workers studied the effects of curing regimes on the shrinkage of Cao-FeOx-Al2O3-SiO2-rich inorganic polymer (IP). After conducted careful experimental study and data analysis, they found that the precursors’ reactivity and curing conditions severely affected IPs shrinkage mechanisms and magnitude. Meanwhile, thermal and moist curing promoted higher volumetric stability and considerably improved IPs mechanical features. The result is meaningful. I think this manuscript can be accept to publish in Materials after careful proofreading to correct the typos.
Reviewer 3 Report
This paper investigates strategies to improve stability of inorganic polymers (namely; CaO-FeOx-Al2O3-SiO2) at ambient and moderate temperatures and conditions. This work also characterized the 28 day formation of pore structures, autogenous 20 shrinkage, drying shrinkage and strength development. This paper is interesting and merits publication once the following items are addressed:
The introduction paints a nice picture of the use of inorganic polymers. The authors may want to improve this discussion by providing few lines covering the use of this material in novel applications such as space construction. https://doi.org/10.2514/6.2009-1015 https://doi.org/10.1016/j.pmatsci.2019.100577 https://doi.org/10.1016/j.asr.2017.06.038 Use “Experiments” and not “Experimental” as a title for Sec. 2.0. Line 133, what dose “additing” refer to? “The samples were sealed with a plastic film during thermal treatment to prevent severe drying.” – why/how would severe drying occur? Figure 10 shows that some materials tend to reduce in compressive strength from 28 days to 56 days. Why is that? How does “Thermal treatment promote a decrease in porosity”?Author Response
Please see the attachment.

Reviewer 4 Report
The subject of the Authors of the article "Shrinkage and mitigation strategies to improve the dimensional stability of CaO-FeOx-Al2O3-SiO2 inorganic polymers" is current and important from both a scientific and an application point of view. Inorganic polymers (IP) materials are much less common than geopolymers (GP). However, the composition of IP materials and the fact that you can synthesize them include from such raw materials as waste (e.g. matallurgical or industrial residues) gives a double environmental benefit and speaks for the need to develop research on this type of composites.
The Authors rightly noted in the summary that the method of curing IP materials at elevated temperature or humidity is practical only for prefabricated elements.
The Authors described in detail the issues related to the preparation and testing of composites. Test results have been subjected to detailed analysis. In Figure 7, it was noted that the Authors drew error bars only for the results obtained after 56 days of puberty - this should be consistently supplemented with other data.
According to the Materials journal guidelines, when formatting literature - vol should be written in italics.
Reviewer 5 Report
The presented manuscript deals with the recent scientific problem in the field of research on reducing shrinkage in building materials. The manuscript is well composed and the topic of the article is valid. Furthermore, the methodology is sufficiently well explained, whereas the results are soundly interpreted and related to existing knowledge. Therefore, the paper is valuable for publication in Materials journal. However, this paper needs revisions and improvements:
(1) In the Introduction, the authors describe the advantages of using Supplementary Cementitious Materials (SCMs). Unfortunately, the Introduction part omitted the characteristics of one of the most commonly used SCMs, i.e. siliceous fly ash. Therefore, these issues should be described and new reports from this topic area cited. For this reason, the following articles should be discussed and cited.
Structural Engineering and Mechanics, vol. 62 (1), 2017, 1-9.
Journal of Civil Engineering and Management, vol. 23 (5), 2017, 613-620.
(2) Please provide more details of experimental program. There is a need to provide the photo of the specimen during fabrication, mixing and curing. Please show also the view of the research stands.
(3) Please show an appearance of the slags used in the research.
(4) Unit designations in tables 4 and 5 should be in brackets.
(5) Please explain exactly why 3 types of samples curing conditions were used. What was the purpose of it ?
(6) Reducing shrinkage in the material should have a positive impact on the improvement of its durability. Please discuss in the article whether the proposed solutions are able to increase the durability of the composites.
Reviewer 6 Report
Dear editor, dear authors,
Thank You for the opportunity to read and review the submitted paper manuscript number materials-632741entitled Shrinkage and mitigation strategies to improve the dimensional stability of CaO-FeOx-Al2O3-SiO2 inorganic polymers.
The scope of the paper is very actual one and it contains valuable data and brings innovative information within the subject. The experiments are proceeded in reasonable way and the obtained results are well discussed and therefore I identify with the conclusions. I have only few comments/answers and suggestions to authors:
Why did you choose the potassium activator? K activators are more expensive and give lower shrinkage compared to Na ones. At the Fig. 1 there is a gap in the particle size distribution curves between 2 and 10 um. Why? What is the PSD measurement method - wet/dry? Please incorporate the method specification into chapter 2.2. I suggest to change the denotation of XRD patterns at Fig. 4 from (1-4) to (1 and 4) etc. This is misleading. Are the volumetric changes at distinct conditions stable if the conditions change later? Have you observed the shrinkage of IPs when removed from the environment of 100% RH? I suggest to add at least short discussion of this topic to the paper. Please revise the manuscript, there are some typing errors. I’ve catch some, e.g.: r.81 “…CaO-FeOx-Al2O3-SiO2-rich slags as been considered…”, r.561 “porous structured formed”The paper is very interesting and I recommend it to accept for publication in Materials after minor corrections.
Round 2
Reviewer 5 Report
I have no comments.